# Changes in Sex Difference in Time-Limited Ultra-Cycling Races from 6 Hours to 24 Hours

**DOI:** 10.3390/medicina57090923

**Published:** 2021-09-01

**Authors:** Heike Scholz, Caio Victor Sousa, Sabrina Baumgartner, Thomas Rosemann, Beat Knechtle

**Affiliations:** 1Institute of Primary Care, University of Zurich, 8091 Zurich, Switzerland; heikescholzkn@gmail.com (H.S.); Thomas.rosemann@usz.ch (T.R.); 2Health Technology Lab, Bouvé College of Health Sciences, Northeastern University, Boston, MA 02115, USA; cvsousa89@gmail.com; 3Medbase St. Gallen Am Vadianplatz, 9001 St. Gallen, Switzerland; Sabrina.Baumgartner@usz.ch

**Keywords:** ultra-endurance, performance, gender

## Abstract

*Background and objective*: Existing research shows that the sex differences in distance-limited ultra-cycling races decreased with both increasing race distance and increasing age. It is unknown, however, whether the sex differences in time-limited ultra-cycling races will equally decrease with increasing race distance and age. This study aimed to examine the sex differences regarding performance for time-limited ultra-cycling races (6, 12, and 24 h). *Methods***:** Data were obtained from the online database of the Ultra-Cycling Marathon Association (UMCA) of time-limited ultra-cycling races (6, 12, and 24 h) from the years 1983–2019. A total of 18,241 race results were analyzed to compare cycling speed between men and women by calendar year, age group (<29; 30–39; 40–49; 50–59; 60–69; >70 years), and race duration. *Results:* The participation of both men (85.1%) and women (14.9%) increased between 1983 and 2019. The age of peak performance was between 40 and 59 years for men and between 30 and 59 years for women. Between 2000 and 2019, more men (63.1% of male participants and 52.2% of female participants) competed in 24 h races. In the 24 h races, the sex difference decreased significantly in all age groups. Men cycled 9.6% faster than women in the 12 h races and 4% faster in the 24 h races. Both women and men improved their performance significantly across the decades. Between 2000 and 2019, the improvement in the 24 h races were 15.6% for men and 21.9% for women. *Conclusion:* The sex differences in cycling speed decreased between men and women with increasing duration of ultra-cycling races and with increasing age. Women showed a greater performance improvement than men in the last 20 years. The average cycling speed of men and women started to converge in the 24 h races.

## 1. Introduction

Over the past 25 years, ultra-endurance events have gained increasing popularity [1,2]. Current trends include athletic events such as Ironman triathlons [3], adventure racing [4], the 3000-mile Race Across America cycling race [5], and Ultraman triathlons [6].

Many of the events have been introduced to appeal to athletes who seek longer-lasting challenges [1]. In ultra-marathon foot races, participation trends increased more than in other ultra-endurance events [2]. The increase in ultra-endurance events such as ultra-cycling [7] and ultra-marathon running [2] can mainly be explained by the larger number of women and master athletes [2] competing in these ultra-endurance events. Generally, female participation in ultra-endurance races such as ultra-marathon running is low [2] but has shown an increasing trend in recent decades [8].

Ultra-endurance competitions are solo events that attempt to challenge the limits of human endurance [9]. Ultra-endurance events are defined as any endurance performance longer than 6 h in duration [2]. For cycling, the Ultra Marathon Cycling Association (UMCA) describes an ultra-cycling race as a race of 100 miles or more, lasting at least six hours [10]. These races can be classified in time-limited (e.g., 6, 12, and 24 h races) and distance-limited (e.g., 100, 200, 400, and 500 miles) ultra-cycling races [10].

The discussion of sex differences is a recurrent topic in sports physiology and has occupied scientists for decades [11,12,13]. One finding was that peripheral fatigue is greater for men than women due to a different distribution of muscle fiber types [11]. The sex differences in endurance and ultra-endurance performances have been investigated for almost 70 years [14,15,16,17]. Initially, in the 1990s, Whipp and Ward speculated that women could outperform men in running [16]. Women showed a better performance improvement because they participated in sports competitions such as the women’s Olympic 800 m, the Boston Marathon, and the women’s Olympic marathon in the 1960s [18]. However, sex differences in sports remain an ongoing topic in science, literature, and media [19]. With the higher participation rate of women in ultra-endurance events (e.g., ultra-marathon running races, ultra-triathlons, ultra-distance swimming, ultra-cycling, cross-country skiing) over the last 25 years [2], the question evolved into whether women would be able to narrow the gap with men in different sports such as ultra-marathon running [20], swimming [21], ultra-trail running [22], and open-water ultra-distance swimming [23,24]. In ultra-cycling, the question has not been answered thoroughly [25].

Different studies showed that the sex differences depend on both the sports modality [26] and the race distance [19]. Concerning endurance sports, the lowest sex difference is in long-distance open-water swimming (distance of ~30 km), especially with water colder than ~20 °C [21]. Even though the interest in investigating the sex differences in ultra-endurance performances such as ultra-triathlon [27], ultra-running [28], and ultra-swimming [21] has increased, little is known about the sex differences in ultra-cycling [7,27,29].

Due to the results regarding the sex differences in performance improvement in swimming [30] and ultra-triathlons [31], we speculated whether the results were similar in time-limited ultra-cycling. For example, in swimming, the performance improvement at the Olympic Games and in the FINA World Championships increased between 1992 and 2013 linearly within both male and female athletes [30]. The sex difference decreased non-linearly, showing that finalists and champions at the Olympic Games and FINA World Championships reduced the sex difference with increasing race distance. Recent studies indicated that women could outperform men in ultra-swimming events [21,32]. In the shorter ultra-triathlon distances (i.e., Ironman distance covering 3.8 km swimming, 180 km cycling, and 42.195 km running), the sex difference increased but decreased in longer distances (i.e., Double and Triple Iron ultra-triathlon) [31]. The conclusion was that women would be unlikely to outperform men in ultra-triathlons from the Ironman to Double Iron ultra-triathlon [31]. Whether the sex differences would decrease or increase linearly or non-linearly over the years is yet to be investigated for time-limited ultra-cycling races.

Due to an increasing number of female and male athletes competing in distance-limited ultra-cycling races over recent decades, the differences in the performance level between the sexes were examined [25]. The study by Baumgartner et al. collected data from the year 1996 to 2018 from the available online database of the Ultra-Cycling Marathon Association (UMCA), including distance-limited ultra-cycling races (100, 200, 400, and 500 miles) [25]. Though women were slower than men over all distances (160 to 800 km) in ultra-cycling [25,27], sex differences in distance-limited ultra-cycling races decreased with increasing race distance and increasing age [25]. The average sex difference in ultra-cycling accounts for ~18–28% regarding an analysis of the 5000 km Race Across America (RAAM) and two RAAM qualifier races, the Furnace Creek 508 (800 km) and the Swiss Cycling Marathon in Europe (715 km) [7]. The sex difference in distance-limited ultra-cycling races was examined and showed a lack of knowledge of whether women can reduce sex differences in time-limited ultra-cycling races. The conditions for every participant in distance-limited races could be very different, while the conditions in time-limited races remain relatively stable. The results in the study by Baumgartner et al. could be affected by outside influences [25]. In time-limited cycling races, outside factors such as wind could mainly be excluded. It is unknown whether the sex differences would decrease with increasing age and race time in time-limited ultra-cycling races. This study will fill the gap in the knowledge on sex differences in time-limited ultra-cycling races.

In distance-limited ultra-cycling races, participants generally overcome a specifically defined distance [25]. In time-limited races such as ultra-marathons, athletes perform laps for a specific period (i.e., 6, 12, 24, 48, 72, 144, and 240 h) [33]. Participants in time-limited ultra-cycling races also ride one lap after the other during a specific period (6, 12, and 24 h) [34]. Men and women have the same competition rules and there are no variations in the circuit. Overall, when we compare time-limited with distance-limited ultra-cycling races, we compare two different race designs. For example, when there is a lot of wind during the race, the tailwind will be neutralized by a headwind in the other half of the lap. From this point of view, we were interested in whether the sex differences would decrease depending on the race duration (hours) and whether this decrease would be more pronounced than the sex differences in distance-limited ultra-cycling races.

Therefore, the present study investigated differences in the performance level of men and women in time-limited ultra-cycling races with the main hypothesis that female ultra-cyclists would reduce the gap with men with increasing race duration and increasing age. The first aim concerning age was to analyze the participation of different age groups in other time-limited races separated by sex. We hypothesized that the period of peak performance would be later in men compared to women. Furthermore, the sex difference would decrease with increasing age. The second aim concerning the sex of the participants was to analyze the differences in participation of the different time-limited races separated by sex. We hypothesized that more men would compete in the longer endurance races (24 h). The third aim concerning the performance level was to analyze the average cycling speed within the longer time-limited races (12 and 24 h) separated by sex. First, we hypothesized that both men and women would improve their performance level over the decades. Second, we hypothesized the average cycling speed would decrease with increasing race duration and that the sex difference would decrease. In other words, women would cycle as fast as men after a specific duration of exertion.

## 2. Materials and Methods

The study analyzed the results of time-limited ultra-cycling races (6, 12, and 24 h) from 1997 till 2019 and compared the performances (average speed) between men and women by calendar year, age group (<29; 30–39; 40–49; 50–59; 60–69; >70 years), and racing time. This study included the average performance of all participants who finished.

### 2.1. Ethical Approval

This study was approved by the Institutional Review Board of Kanton St. Gallen, Switzerland, with a waiver of the requirement for informed consent given that this study analyzed publicly available data (EKSG 01-10-2010).

### 2.2. Data Sampling

The dataset from this study was obtained from the publicly available data of the World Ultra Cycling Association WUCA [10], which is a global, non-profit organization setting the standards for excellence and accomplishment in ultra-cycling. We used the calendar of the Ultramarathon Cycling Association [35] to include the standard races in ultra-cycling worldwide, which indicate sex, age, and race time. Data were collected from the official race websites for 6, 12, and 24 h bike challenges from 1983 to 2019.

The dataset included 33 different events (UMCA) [10] and a total of 18,187 athletes. Required information in the original dataset was the year of competition, sex, name of the athlete, age group, and achieved kilometers in a given time. Based on the different race distances, average cycling speed (km/h) was calculated and used as the primary and dependent variables. The whole dataset was cleaned for double coding results. Much of the European data had to be removed because of missing necessary information, such as age, sex, or distance. For example, known races, such as the Slovenia 6, 12, and 24 h ultra-cycling races (Slo 6-12-24 Ultra), Ultra-Cycling Italy 6, 12, and 24 h, and Road Time Trials Council (RTTC) National Championships, United Kingdom (U.K.), had to be excluded. Data were excluded because of a vague age group. We do not distinguish between nationalities in this study.

### 2.3. Statistical Analysis

The normality and homogeneity of the data were tested using Kolmogorov–Smirnov and Levene’s tests, respectively. The outcome was the average cycling speed (km/h). Descriptive statistics were presented as absolute values (participation outcomes) and means and standard deviations (performance outcomes). Different general linear models were applied as follows: the 1st model was a two-way ANOVA (duration × sex); the 2nd model was a two-way ANOVA (age group × duration), with separate models by sex; the 3rd model was a two-way ANOVA (decade × sex), with different models by duration. The factor “decade” comprehended up to four groups of calendar years: 1983–1989, 1990–1999, 2000–2009, and 2010–2019. A 4th model was applied with “year” as a random factor with each year as a sublevel (1983 to 2019). “Sex” and “age group” were also included as factors for the 4th model. The factor “sex” was always included as a fixed factor, and all other factors were included as random factors. We applied the least significant difference (LSD) post hoc technique for pairwise comparisons when at least one of the main effects was significant. All statistical analyses were carried out with Statistical Software for the Social Sciences (IBM^®^ SPSS v.25, Chicago, IL, USA).

## 3. Results

The final sample included 15,468 (85.1%) men and 2714 (14.9%) women. Table 1 summarizes all considered races.

Participation of both sexes increased over the decades, with the 24 h races starting in 1983 and with more participants over all decades for both men (Figure 1A) and women (Figure 1B). Especially, the participation of women in the 24 h races had tripled over the decades, whereas the participation of men in the 24 h races remained generally stable (Figure 1A,B). The participation of men was the highest in the age group 40–49 years and the 24 h races (Figure 1C). The age of peak performance for men in the 6 and 12 h races was 50–59 years (Figure 1C). For women, most participants were between 30 and 49 years old and competed in the 24 h races (Figure 1D). The age of peak performance in the 6 and 12 h races was between 40 and 59 years (Figure 1D). For men, the fastest athletes were between 40 and 59 years old (Figure 2B). The fastest female athletes competing in 6 and 12 h races were in the age group 30–39 years but for 24 h races, they were in the age group 50–59 years (Figure 2C). The number of female participants in the 6 h races is significantly lower than in the 12 and 24 h races and could not deliver enough data for meaningful results (Figure 1D).

A total of 10,367 men competed between the year 2000 and the year 2019 (Figure 1A). Of them, 6.4% competed in the 6 h races, 30.4% in the 12 h races, and 63.1% in the 24 h races. Between 2000 and 2019, 1995 women participated in time-limited ultra-cycling races (Figure 1B). Of them, 11.6% competed in the 6 h races, 36.2% in the 12 h races, and 52.2% in the 24 h races.

The first general linear model (GLM) showed a significant effect for duration (F = 62.3; *p* = 0.016) and interaction (F = 18.5; *p* < 0.001), but not for sex (F = 7.6; *p* = 0.103). Pairwise comparisons showed that men were faster in all durations, and the average cycling speed significantly decreased with increasing race duration (Figure 2A). The second GLM in men showed significant effects for duration (F = 307.7; *p* < 0.001), age group (F = 7.8; *p* = 0.001), and interaction (F = 4.3; *p* < 0.001). For women, significant effects were found for duration (F = 48.7; *p* < 0.001) and interaction (F = 3.8; *p* < 0.001), but not for age group (F = 2.1; *p* = 0.127). Pairwise comparisons showed for men that the fastest athletes were between 40 and 59 years old and that those athletes aged ≥70 years differed across race durations (Figure 2B). For women, pairwise comparisons showed that the fastest athletes competing in 6 and 12 h races were in the age group 30–39 years but for 24 h races, they were in the age group 50–59 years (Figure 2C). As shown in Figure 2B,C, the average cycling speed of men was significantly higher than those of women in the 6 and 12 h races and for the age groups <29 to 59 years (for men, 26.3 km/h in the 6 h races and 23.4 km/h in the 12 h races; for women, 23.3 km/h in the 6 h races and 20.8 km/h in the 12 h races). In age groups 60–69 and >70 years, the average cycling speed of men increased, and the gap between men and women narrowed (Figure 2B,C) (for men, 22.9 km/h in the 6 h races and 21.2 km/h in the 12 h races; for women in the 6 h races, 22.6 km/h and in the 12 h races, 20.5 km/h). In the 24 h races, the sex difference increased significantly in all age groups (Figure 2A; average cycling speed of men was 16.4 km/h and for women, 15.7 km/h), but the most in the age group <29 to 59 years.

The average cycling speed of men in the 6 h races was 26.2 km/h and 23.5 km/h for women (Figure 2A). Men were 10% faster than women in the 6 h races. In the 12 h races, men cycled with an average speed of 23.2 km/h and women at 21 km/h (Figure 2A). Men cycled 9.6% faster than women in the 12 h races. In the 24 h races, the average cycling speed of men was 16.4 km/h, and for women, 15.7 km/h (Figure 2A). In the 24 h races, men cycled 4% faster than women.

The third GLM showed for 6 h races a significant effect for sex (F = 300.6; *p* = 0.037), but not for decade (F = 52.8; *p* = 0.087) or interaction (F = 0.04; *p* = 0.852). Pairwise comparisons showed that men were faster only in the decade 2010–2019 (Figure 3A). Similarly, for 12 h races, we found a significant effect for sex (F = 26.3; *p* < 0.001), but not for decade (F = 12.3; *p* = 0.075) or interaction (F = 0.77; *p* = 0.465). Pairwise comparisons showed that men were faster in the decades 2000–2009 and 2010–2019 (Figure 3B). Finally, in the 24 h races, we found a significant effect for sex (F = 17.1; *p* = 0.022) and decade (F = 131.5; *p* = 0.001), but not for interaction (F = 1.6; *p* = 0.188). Pairwise comparisons showed that men were faster in all decades, except in 1990–1999. In addition, performance significantly increased over the decades in both men and women (Figure 3C).

For the 6 h races between 2000 and 2019, men showed an improvement of 4% in average cycling speed, and the improvement in women was 5.9% (Figure 3A). In the 12 h races between 2000 and 2019, men improved their performance by 0.4% and women by 3.9% (Figure 3B). Between 2000 and 2019, the improvement in the 24 h races was 15.6% for men and 21.9% for women. The 4th model showed a significant effect for sex (F = 12.1; *p* = 0.002), year (F = 21.0; *p* < 0.001), and age group (F = 9.3; *p* = 0.006) but not for full interaction among all three factors (F = 1.1; *p* = 0.305).

## 4. Discussion

This study intended to investigate the sex difference in ultra-cycling performances for time-limited ultra-cycling races (e.g., 6, 12, and 24 h) with the hypothesis that the sex difference would decrease with increasing age and increasing race duration. The main findings were (i) the overall participation of men in time-limited ultra-cycling races was higher than for women with higher participation of men in the longer endurance races, (ii) the age of peak performance of men in time-limited ultra-cycling races was higher than that in women, (iii) the sex difference in performance decreased with increasing age, (iv) cycling speed decreased with increasing race duration, (v) the sex difference in cycling speed decreased with increasing race duration, and (vi) men and women improved their performance over the years.

### 4.1. Decreasing Sex Difference with Increasing Race Duration

The main finding was a decrease in cycling speed and a reduction in sex difference with increasing race duration. The cycling speed of men decreased more than that of women, confirming our hypothesis. An investigation of performance trends in the “Swiss Cycling Marathon” (110 to 900 km) showed that sex differences decreased over past years (2001 to 2012), reaching ~14% in 2012 [29]. Recently, a retrospective data analysis of distance-limited ultra-cycling races (100, 200, 400, and 500 miles), including 12,716 race results from the year 1996 to the year 2018, showed a narrowing sex gap with increasing distance and age [25]. The sex gap was the widest in the 100-mile races and narrowed for longer distances such as the 200-, 400-, and 500-mile races [25]. A potential explanation may lie in women’s anthropometric and physiological characteristics, such as a higher percentage of body fat, which stores energy [36]. Another cause is that the anaerobic energy supply (preserved by the impact of testosterone) is subordinated in ultra-endurance sports [25]. This may explain why women can cycle as fast as men in the 24 h races.

### 4.2. Sex Differences Concerning the Age and Biological Requirements of the Participants

Another important finding was that the fastest men in time-limited ultra-cycling races were between 40 and 59 years old, whereas the fastest women were between 30 and 59. As anticipated, the age of peak performance was higher in men compared to women. The age of peak performance in ultra-cycling is generally higher [7] than those in other sports, such as marathon running (25–35 years), baseball (28 years), or tennis (24 years) [37]. Most of these older participants in ultra-cycling events were master athletes [2]. Age-related changes in performance could be compensated due to an increased pre-race experience [38,39] and more training hours [40].

In both time-limited and distance-limited ultra-cycling races, the sex difference in the performance level decreased with increasing age [25]. This effect is also documented in ultra-marathon running [41] and swimming disciplines such as long-distance open-water swimming [21]. Potential explanations could be found in the sex difference concerning fatigue tolerance [42] and the change in muscle fiber recruitments with increasing age [43]. Peripheral fatigue is greater for men than women due to a different distribution of muscle fiber types [11]. Except for the sex difference in muscle fiber size, there seem to be no differences in muscle tissue “quality” between the sexes [13]. However, generally, women have a lower total skeletal muscle (SM) mass/fat free mass (FFM) ratio than men [44].

### 4.3. Sex Differences in the Participation

As we suggested, the participation of men in time-limited ultra-cycling events was higher than for women in all disciplines. The participation of women in ultra-cycling events ranged from 11% in the RAAM and the Furnace Creek 508 to 3% in the Swiss Cycling Marathon [2,7]. In distance-limited ultra-cycling races, the participation of women over the years in all distances was 9% [25]. The differences in female participation over the years can be explained by the fact that female participation for different race designs started in different years. For example, women were first allowed to run in the Olympic Marathon in 1984 [45]. In 1972, female participants were accepted to run the Boston Marathon [46].

Social and psychological reasons can most likely explain the overall lower participation of women in ultra-cycling events as for ultra-runners [47]. It can be challenging to balance training hours with work, relationships, and parenting [47]. The motivation of female ultra-runners was a personal achievement and general health, whereas social recognition played a subordinate role as a motive for running [47]. Another important reason for the sex difference in participation might be psychological motives. Exercise is often used to compensate for stress, as a coping mechanism, to find inner peace, and to build self-esteem [48]. Due to existing sex differences in psychological variables, such as competitive anxiety, emotional stability, or emotion control [48], it is important to include psychological motives, which may lead to the assumption that women tend to choose the less challenging races for different motivational reasons.

More men joined the longer-lasting cycling races (24 h) instead of the shorter ones than women. Due to the study by Shoak et al. [7], which came to similar conclusions on the participation, we anticipated that men are more likely than women to join the longer-lasting challenges. The investigation of participation and performance trends in ultra-cycling showed that in races longer than the marathon distance, female participants decreased with increasing race distance [7]. An explanation could be found in the sex difference in training motivation in runners for fast running performances [49]. Men are more often motivated in dedicated, high-volume training, which is necessary for fast running performances [49]. This sex difference in training motivation could indicate the male predisposition for enduring competitiveness [50]. In that case, distance running may function as an indicator of quality to others, and enduring competitiveness may serve as a display for this quality [50]. The long-term achievement motivation of men often takes place in display or “show-off” domains. In contrast, the long-term achievement motivation of women takes place in domains more relevant to resource acquisition [50]. Therefore, men could join the longer-lasting races (24 h) for competitive reasons. Further studies need to investigate the sex differences in motivation for ultra-endurance cyclists.

### 4.4. Performance Improvement

The last finding was that both men and women showed a performance improvement over the decades. The data confirm our hypothesis that men and women would improve their performance over the years. The long duration and the high intensity of an ultra-endurance race create physiological challenges [51]. Therefore, athletes require fluid and energy replacement to control body temperature and sustain endurance [52]. The high caloric expenditures often are compensated with substrate utilization to ensure the supply of carbohydrates, fat, and protein [52]. Commercial sports drinks containing maltodextrin can serve as a source of carbohydrate [52]. The research for optimal nutrition during ultra-endurance events decreased over the years [53,54] and enabled athletes to improve their performance.

Another reason for the performance improvement is the participation of master athletes spending more time on training [40] and showing more pre-race experience [39]. A psychological study showed the positive influence of a successful pre-race experience on motivation [50].

Technological developments improved the performance of athletes, such as the Schoberer Rad Messtechnik (SRM) PowerMeter, which was invented 1986 by Ulrich Schoberer [55]. SRM made it possible for cyclists to measure the performance while riding under real conditions [55]. Until 1986, there was no adequate method to measure the performance while participating in a competition, only ergometer tests in the laboratory [55].

Another example of an invention is the first aerodynamic spoked wheel made entirely of fiber composite material, which Rudolf Dierl and Heinz Obermayer developed in 1995 [56]. These “lightweight” carbon wheels combined low weight and good aerodynamics for optimal performance [56,57]. The last example is the improvement of aerodynamics in cycling. The reduction in air resistance gives the possibility to become faster [58]. Special time trial bikes, aero bikes, aero bars, skinsuits, and aero helmets [59] are reducing air resistance as much as possible [58,60].

Over the years, the performance of men and women in different sports disciplines has been improved by exercise testing in sports medicine [61]. One example for the development in exercise through sports chemistry is low-level laser therapy (LLLT) [62]. LLLT delayed the development of skeletal muscle fatigue and concurrently decreased postexercise levels of biochemical markers of muscle recovery [62]. It could be helpful to delay fatigue during a repetitive task and help recovery [62]. Investigations studying the effect of recovery modalities, such as cryotherapy, contrast temperature water immersion therapy, hyperbaric oxygen therapy, nonsteroidal anti-inflammatory drugs, and compression garments, could also lead to performance improvement in athletes [63].

In conclusion, the performance improvement in men and women could be explained by the general improvements through scientific research in different areas. Women showed a more significant improvement than men in average cycling speed over the decades. Further studies investigating the reasons for the larger performance improvement in women are needed.

### 4.5. Limitations, Strengths, Weaknesses, and Practical Applications

A limitation of this study was the missing data concerning the participation in the 6 and 12 h races in earlier decades. For the 12 h races, data were available from the year 1990 to 2019. For the 6 h races, data were available from the year 2000 to 2019. Therefore, we could only use the data from the year 2000 to 2019 for comparisons. Another limitation was the small number of female participants, especially in the 6 and 12 h races, above 59 years of age. A strength of this study was the extensive dataset containing 33 different events and a total of 18,187 athletes. For practical applications, coaches and ultra-cyclers might set long-term training goals considering their sex and age. For example, considering the decreasing performance gap between men and women with increasing racing time, women could be encouraged to compete in the longer-lasting, time-limited ultra-cycling races. Additionally, elderly athletes could be encouraged to choose longer-lasting events because of the importance of experience. As our study showed, pre-race experience could influence motivation and lead to better performances no matter the participant’s age.

## 5. Conclusions

The existing sex differences between men and women in cycling speed decreased with the duration of ultra-cycling races and with increasing age. Women cycled nearly as fast as men in the 24 h races. The performance of men and women could be equalized because of the greater improvement of women in cycling speed over the decades in time-limited ultra-cycling races. The improvement is shown in all age groups. The average speed of men and women started to converge in the 24 h races. With the results of this study, cyclists, sports medicine specialists, and coaches could set long-term training goals. It would be of interest to examine the reasons for the greater performance improvement in women and the reasons for the better performances of women in longer lasting ultra-cycling races.

## Figures and Tables

**Figure 1 medicina-57-00923-f001:**
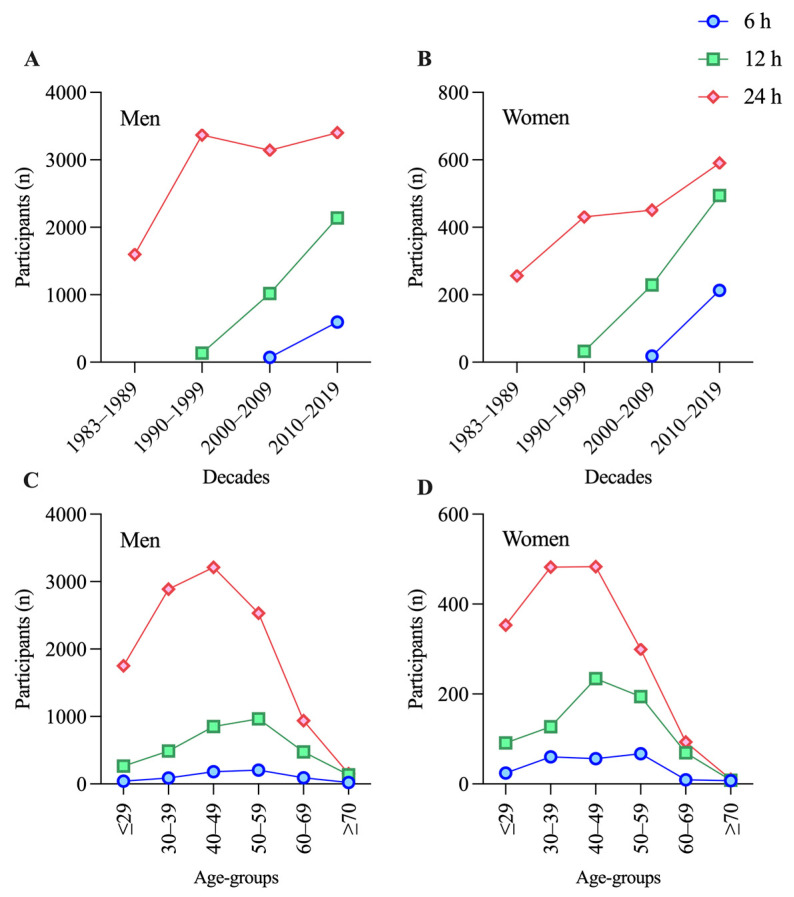
Participation of men and women in 6, 12, and 24 h races. (**A**) Men divided by decade; (**B**) Women divided by decade; (**C**) Men divided by age-group; (**D**) Women divided by age-group.

**Figure 2 medicina-57-00923-f002:**
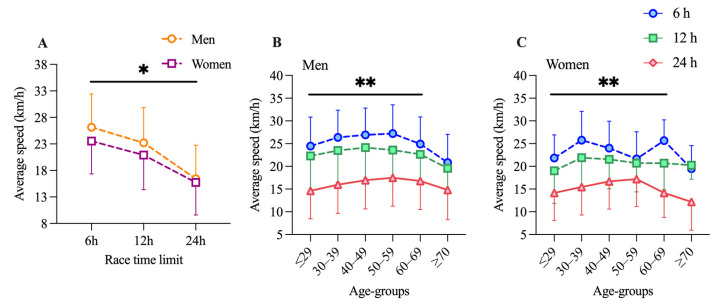
Average speed (km/h) of men and women (**A**). Average speed (km/h) of men (**B**) and women (**C**) subdivided into 6, 12, and 24 h races and age groups. Data expressed as mean and standard deviation. * difference between men and women for each distance below the line. ** difference between race distances for each age group below the line.

**Figure 3 medicina-57-00923-f003:**
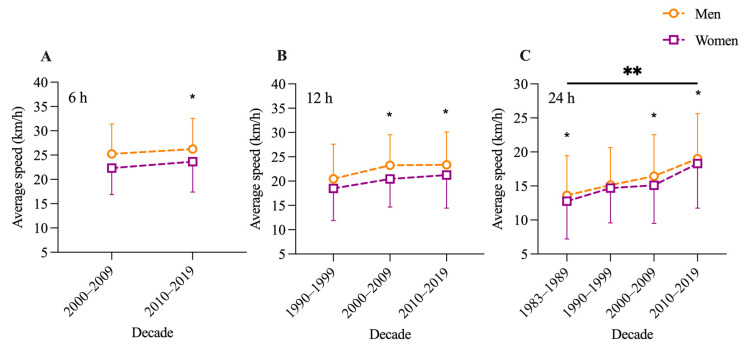
Average speed (km/h) of men and women, subdivided into 6 (**A**), 12 (**B**) and 24 h (**C**) races and decade. Data expressed as mean and standard deviation. *: difference between sexes within specific decade. **: difference between each decade.

**Table 1 medicina-57-00923-t001:** All included time-limited ultra-cycling races.

Time-Limited Races	Considered Editions	Number of Years	*n* (Women)	*n* (Men)	*n* (Total)
6 h Limited Races					
24 h in the Canyon	2014–2018	5	16	18	34
Bessies Creek	2012–2018	7	20	28	48
Calvin Challenge	2008–2018	11	50	213	263
FatAss	2014–2019	6	10	33	43
Maryland Endurance Challenge	2017–2019	3	11	67	78
Pace Bend Ultra	2018	1	6	5	11
Pace Bend	2017–2018	2	11	17	28
Texas Time Trials	2005–2019	15	52	176	228
Texas Ultra Spirit	2017	1	3	4	7
World Time Trial Championship	2012–2019	8	53	106	159
12 h Limited Races					
24 h in the Canyon	2018–2019	2	8	5	13
Bessies Creek 12 h	2012–2018	7	14	89	103
Big Jay’s Fat Ass Race in Illinois	2014–2019	6	49	153	202
Bike Sebring 12 h	2005–2019	15	140	593	733
Calvin Challenge	1997–2018	22	340	1638	1978
Ring of Fire T.T.	2007–2011	5	5	0	5
Hudson River Ramble 12	2004–2018	15	29	152	181
Lewis and Clark	2012	1	2	9	11
Maryland Endurance Challenge	2019	1	13	52	65
Mid Atlantic	2012–2019	8	18	118	136
Texas Time Trial	2004–2019	16	59	189	248
World Time Trial Championship	2011–2019	9	43	149	192
XtremeMelfar	2015–2017	3	34	143	177
24 h Limited Races					
24 h in the Canyon	2014–2018	5	19	53	72
Bessies Creek	2012–2018	7	12	71	83
Bike Sebring 24 h	2006–2019	14	97	582	679
World Time Trial Championship	2011–2018	8	48	226	274
Mid Atlantic	2012–2019	8	8	103	111
National 24 h	1983–2019	37	1405	9759	11,164
Saratoga Challenge 24	1994–2018	25	16	172	188
Texas Time Trials	2004–2019	26	38	181	219
World Championships	2011–2019	9	66	314	380
Mersey Roads Club 24 h	2019	1	2	6	8
Xtreme Melfar	2015–2016	2	17	44	61

## Data Availability

Available online: www.ultracycling.com/calendar-overview/.

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
