# Peer review of "Changes in Sex Difference in Time-Limited Ultra-Cycling Races from 6 Hours to 24 Hours"

_medicina, 2021, doi:10.3390/medicina57090923_

Round 1
Reviewer 1 Report
The reviewed paper is a very insightful analysis of athletic performance in endurance sports over the period 1983-2019. Unfortunately, it has serious shortcomings, which the authors should complete and explain.
- The Author received approval from the committee for the study in 2010- why such a long period between the application and submission of the paper for publication.
- Due to analysis of available literature and internet data - what was the main topic of the application to the ethics committee.
- Very precise and clear presentation of the methodology of literature analysis and obtained research results.
- The authors did not explain in the discussion the background of obtained research results or postulated trends. The physiological and anatomical background of the response to training stimuli of men and women was not explained. No mention was made of the difference in fatigue tolerance between genders or the change in e.g. muscle fibre recruitment with age and the loss of performance is only explained by sarcopenia.
- Conclusion about the increased participation of women in sport is obvious over the years, in probably every sport discipline.
- According to the authors, the increase of athletes' efficiency is mainly influenced by the development of technology. Please mention the importance of exercise testing in the assessment of performance and predicting the training load. Development of sports biochemistry, sports dietetics or pharmacological support and development of wellness systems.
- The journal to which the paper is submitted includes publications on medical topics or the explanation of occurring phenomena is based on physiological-biochemical analysis. The work presented for review is rather of reporting nature and fits into the typical sports journals.
- Literature record is inconsistent with the requirements of the journal; to be corrected
Sincerely,
Author Response
Reviewer 1
The reviewed paper is a very insightful analysis of athletic performance in endurance sports over the period 1983-2019. Unfortunately, it has serious shortcomings, which the authors should complete and explain.
The Author received approval from the committee for the study in 2010- why such a long period between the application and submission of the paper for publication.
Answer: We agree with the expert reviewer. The reason for the long period between the application and submission can be found in the missing data regarding female participation in time-limited ultra-cycling races. As we stated in our study the data between the year 1983 and 2000 is not solid because of the missing female participation. Furthermore, there were not as much different races and the data were not collected as precise as in the years after 2000 (e.g., age, sex, lap rounds).
Due to analysis of available literature and internet data - what was the main topic of the application to the ethics committee.
Answer: We didn`t need an ethic permission for this kind of study.
Very precise and clear presentation of the methodology of literature analysis and obtained research results.
The authors did not explain in the discussion the background of obtained research results or postulated trends. The physiological and anatomical background of the response to training stimuli of men and women was not explained. No mention was made of the difference in fatigue tolerance between genders or the change in e.g., muscle fibre recruitment with age and the loss of performance is only explained by sarcopenia.
Answer: We choose to mention only one factor for the decreasing sex differences with increasing age because we wanted to have a clear structure. For the interested readers there are other studies dealing with physiological explanations for the decreasing sex differences with increasing age.
Conclusion about the increased participation of women in sport is obvious over the years, in probably every sport discipline.
Answer: We agree with the expert reviewer but in our opinion, it is important to mention the reasons for the increasing participation of women. With the higher participation of women, the data become more valid. Therefore, it is important to mention the increased participation of women. Furthermore, the participation increased in nearly every sport discipline, but decreased with the longer lasting disciplines. This trend is important to mention.
According to the authors, the increase of athletes' efficiency is mainly influenced by the development of technology. Please mention the importance of exercise testing in the assessment of performance and predicting the training load. Development of sports biochemistry, sports dietetics or pharmacological support and development of wellness systems.
Answer: We explained the performance improvement with different reasons (technological and diet-related improvement, pre-race experience, more training hours…). We choose to pick some examples for technological improvements. More reasons for the performance improvement would go beyond the scope, but we will add another example for exercise testing.
The journal to which the paper is submitted includes publications on medical topics or the explanation of occurring phenomena is based on physiological-biochemical analysis. The work presented for review is rather of reporting nature and fits into the typical sports journals.
Answer: The differences in performance between the sexes is biologically conditioned. In our study we showed the dependency of performance on the load duration. The results of this study are of interest for physiologists to search for a reason for the better performances of women in longer lasting challenges.
Literature record is inconsistent with the requirements of the journal; to be corrected
Answer: We agree with the expert reviewer and changed as suggested.
Reviewer 2 Report
Dear Authors,
The manuscript is of a really high standard. I read the text twice and from my perspective the authors approached the presentation of their research very honestly and seriously. It is practically difficult to find shortcomings.
From my perspective, only the final conclusions should be expanded.
What do this text and research bring to contemporary literature?
What tips do we have for cyclists?
Who has gained the most and will use the results of your research?
Author Response
Reviewer 2
Dear Authors,
The manuscript is of a really high standard. I read the text twice and from my perspective the authors approached the presentation of their research very honestly and seriously. It is practically difficult to find shortcomings.
From my perspective, only the final conclusions should be expanded.
Answer: We agree with the expert reviewer and changed to ‘The existing sex differences between men and women in cycling speed decreased with the duration of ultra-cycling races and with increasing age. Women cycled nearly as fast as men in the 24-hour races. The performance between men and women could be equalized because of the greater improvement of women in cycling speed during the decades time-limited ultra-cycling races. The improvement is shown in all age groups. The average speed of men and women started to converge in the 24-hour races. With the results of this study cyclists, sports medicine specialists and coaches could set long-term training goals. It would be of interest to examine the reasons for the greater performance improvement in women and the reasons for the better performances of women in longer lasting ultra-cycling races.
What do this text and research bring to contemporary literature?
Answer: Our study makes a contribution to the exploration of sex differences in ultra-endurance sports. There are studies which investigate the sex differences in swimming, running and ultra-triathlon, but our study is the first study concerning sex differences in time-limited ultra-cycling races.
What tips do we have for cyclists?
Answer: We stated practical applications in the discussion under 4.5. For example, ultra-cyclers might set long-term training goals considering their sex and age. Women could be encouraged to compete in the longer lasting time-limited ultra-cycling races. Also, elderly athletes could be encouraged to choose longer lasting events because of the importance of experience. Pre-race experience could influence motivation.
Who has gained the most and will use the results of your research?
Answer: Cyclists, sports medicine specialists and coaches could use the results of our study.
Reviewer 3 Report
I am grateful to have read the article entitled "Female ultra-cyclists reduce the gap to men with increasing race duration and increasing age" which aims to "examine the sex differences regarding performance for time-limited ultra-cycling races". Sex differences in sporting performance, certainly a topic that remains interesting despite the many existing publications on the topic. The reading of the article has been difficult for me, it seems to me that in the introduction there is an excess of information, in a certain way reiterated, not essential for the objectives of the work, which makes the reading difficult. It could be expressed in a more summarized and orderly manner, and these references could be used in greater depth in the discussion. The approach, development and conclusions of the article are very similar to other articles already published, for example the one in your reference 25. They should focus more on trying to highlight the originality of their study. It is not a review article, so I advise you to reduce and be more selective in the choice of bibliographic references and in the arguments; for example, is it necessary to refer to whether sarcopenia is more or less frequent in men than in women? (44), do they believe that a sarcopenic person would be able to compete in utra-endurance competitions?
The assertion that "Different studies show that the sex differences depend on both the sports modality", needs to be supported by reference 26. is it necessary to support with reference 26, is this reference important? Even in this reference it speaks of differences between sexes depending on the type of sport? I think not (https://journals.humankinetics.com/view/journals/wspaj/18/2/article-p76.xml)
I would encourage you to rewrite using "not tangential" references, but really valid, relevant and meaningful references; and to carefully review the bibliography you wish to use; for example, (and it is just an example), reference 19 is not from 2017, but from 2018, (https://pubmed.ncbi.nlm.nih.gov/29466055/ )
I think that the objectives should be synthesized more, and I do not think it is necessary to establish a hypothesis for each subsection (especially when some of these objectives have already been answered in other studies and do not pose particularly novel hypotheses), I think they should only make a more generic hypothesis, because in the discussion the use of "confirming our hypothesis" (lin.:274, 329= or "confirming our hypothesis" (lin.:260, 292, 312) is very reiterative.
I believe that it has important methodological inaccuracies or errors in order to reach a valid conclusion. Thus an important aspect, not clarified in the article, is whether the comparisons between men and women in the same races have been made only between male winners versus female winners or have used the average performance of all athletes who finished each of the tests by age and gender. If only the time of the winners was used, it would be better to use the expression "difference between male and female winners" rather than "male and female differences" throughout the text. It would certainly be more illustrative to study the average performance of all participants by age group and gender.
On the other hand, they should clarify to what extent in the different years collected, the winner has been repeated, otherwise there will be an important bias of repeatedly making the same comparison between the same man and the same woman, so that the differences that could be found should not be extrapolated to all men and women. This is necessary to better understand the relevance of the statistical procedure used for comparison over time.
Another aspect that should be clarified is whether the circuits and competition rules are the same between the same test for men and women, as they are often performed with variations in the circuit that influences the final time or distance.
Since we are trying to see the evolution in the difference in performance over the years as a function of test time, I suggest that you employ the same year dimensioning for all three tests, and not one from 2000 to 2019, one from 1990 to 2019, and one from 1983 to 2019. All from 2000 to 2019.
Have you conducted a study of the normality of the distribution of the variables? In the comparison either by age group or by gender, did they use any type of correction when the variances between the groups were different? Did they perform post hoc tests? What were the results and what were the results?
With what statistical tests did you compare the trend in improvement between men and women, to state (Lin 355-356) "Though women showed a more significant improvement than men in average cycling speed over the decades, the gap to men couldn't be closed but narrowed."? ) (Lin. 355-7)
At the bottom of the figures, the significance of the symbols used (*, **, bold horizontal line, ....) should be specified. The error bars in Figures 2 and 3 appear to have the same magnitude within each subfigure (a, b c); what do the error bars in the graphs represent: standard deviation, standard error of the mean?). I encourage the authors to offer as supplementary material the data shown and used in their analysis.
With which statistical tests were the pairwise comparisons performed (Lin. 22) to state: "Pairwise comparisons showed 2 that men were faster in all decades"?
In the statistical section, they should describe in detail all the statistical tests used, including post hoc tests.
Conclusions should be drawn on the basis of the data shown:
With the data shown you cannot state in the conclusion: "The greater improvement of women in cycling speed during the decades supports the hypothesis that women could outperform men in time-limited ultra-cycling races", perhaps you could state that the performances between men and women could be equalized.
Author Response
Reviewer 3
I am grateful to have read the article entitled "Female ultra-cyclists reduce the gap to men with increasing race duration and increasing age" which aims to "examine the sex differences regarding performance for time-limited ultra-cycling races". Sex differences in sporting performance, certainly a topic that remains interesting despite the many existing publications on the topic.
The reading of the article has been difficult for me, it seems to me that in the introduction there is an excess of information, in a certain way reiterated, not essential for the objectives of the work, which makes the reading difficult. It could be expressed in a more summarized and orderly manner, and these references could be used in greater depth in the discussion.
Answer: The structuring of our study is necessary to introduce into the topic. In the first part we describe the popularity of ultra-endurance events, then we state the description of such events. Then we introduce into the discussion of sex differences as an ongoing topic in sports medicine. As a last step we showed the state of research in other sports disciplines to show the lack of research in time-limited ultra-cycling races.
The approach, development and conclusions of the article are very similar to other articles already published, for example the one in your reference 25. They should focus more on trying to highlight the originality of their study. It is not a review article, so I advise you to reduce and be more selective in the choice of bibliographic references and in the arguments; for example, is it necessary to refer to whether sarcopenia is more or less frequent in men than in women? (44), do they believe that a sarcopenic person would be able to compete in ultra-endurance competitions?
Answer: We agree with the expert reviewer and there are many studies examining sex differences in sports. But this is the first study examining sex differences in time-limited ultra-cycling races, which is a different race design than distance-limited ultra-cycling races. Time-limited ultra-cycling races are less susceptible to external influences. Regarding sarcopenia, it is important to describe the performance differences between men and women with increasing age. Sarcopenia could be one explanation for the equalization in female performances to men with increasing age.
The assertion that "Different studies show that the sex differences depend on both the sports modality", needs to be supported by reference 26. is it necessary to support with reference 26, is this reference important? Even in this reference it speaks of differences between sexes depending on the type of sport? I think not (https://journals.humankinetics.com/view/journals/wspaj/18/2/article-p76.xml)
Answer: We agree with the expert reviewer and removed the reference 26 from the study. It is not important for our study.
I would encourage you to rewrite using "not tangential" references, but really valid, relevant and meaningful references; and to carefully review the bibliography you wish to use; for example, (and it is just an example), reference 19 is not from 2017, but from 2018, (https://pubmed.ncbi.nlm.nih.gov/29466055/)
Answer: We agree with the expert reviewer and corrected the bibliography.
I think that the objectives should be synthesized more, and I do not think it is necessary to establish a hypothesis for each subsection (especially when some of these objectives have already been answered in other studies and do not pose particularly novel hypotheses), I think they should only make a more generic hypothesis, because in the discussion the use of "confirming our hypothesis" (lin.:274, 329= or "confirming our hypothesis" (lin.:260, 292, 312) is very reiterative.
Answer: We agree with the expert reviewer and completed the main hypothesis, but it is important to clarify the other important results via hypothesizes for each subsection.
I believe that it has important methodological inaccuracies or errors in order to reach a valid conclusion. Thus an important aspect, not clarified in the article, is whether the comparisons between men and women in the same races have been made only between male winners versus female winners or have used the average performance of all athletes who finished each of the tests by age and gender. If only the time of the winners was used, it would be better to use the expression "difference between male and female winners" rather than "male and female differences" throughout the text. It would certainly be more illustrative to study the average performance of all participants by age group and gender.
Answer: We used the average performance of all athletes by age and gender. We will correct it in the text.
On the other hand, they should clarify to what extent in the different years collected, the winner has been repeated, otherwise there will be an important bias of repeatedly making the same comparison between the same man and the same woman, so that the differences that could be found should not be extrapolated to all men and women. This is necessary to better understand the relevance of the statistical procedure used for comparison over time.
Answer: We did not only include the winners for this study, but all the participants who finished. Therefore, there is not the same comparison between one man and one woman. We completed it in the methods.
Another aspect that should be clarified is whether the circuits and competition rules are the same between the same test for men and women, as they are often performed with variations in the circuit that influences the final time or distance.
Answer: Men and women have the same competition rules and there are no variations in the circuit. The cyclists drive one lap after another no longer than about 3- 10 miles. We will complete the description of time-limited ultra-cycling races in the text.
Since we are trying to see the evolution in the difference in performance over the years as a function of test time, I suggest that you employ the same year dimensioning for all three tests, and not one from 2000 to 2019, one from 1990 to 2019, and one from 1983 to 2019. All from 2000 to 2019.
Answer: We agree with the expert reviewer. The reason for grouping the years is because we have other two important factors as potential sub-groups (age-group and sex), and some data can end up being excluded from the analysis because they do not have variance with zero or one participant (i.e., 1983, men, 40-49: n = 1). Nevertheless, we did run the model as the reviewer suggested and added it as text to the results section.
Have you conducted a study of the normality of the distribution of the variables? In the comparison either by age group or by gender, did they use any type of correction when the variances between the groups were different? Did they perform post hoc tests? What were the results and what were the results?
Answer: The normality and homogeneity of data were tested using Kolmogorov-Smirnov and Levene’s tests, respectively. All data showed normal, homogeneous, or both, thus, parametric statistical tests were applied. The post-hoc test we used was the least significant difference (LSD). All this information was added to the Statistical Analysis section, as suggested in the next comments.
With what statistical tests did you compare the trend in improvement between men and women, to state (Lin 355-356) "Though women showed a more significant improvement than men in average cycling speed over the decades, the gap to men couldn't be closed but narrowed."? ) (Lin. 355-7)
Answer: We agree with the expert reviewer and changed as requested.
At the bottom of the figures, the significance of the symbols used (*, **, bold horizontal line, ....) should be specified. The error bars in Figures 2 and 3 appear to have the same magnitude within each subfigure (a, b c); what do the error bars in the graphs represent: standard deviation, standard error of the mean?). I encourage the authors to offer as supplementary material the data shown and used in their analysis.
Answer: We agree with the expert reviewer. The figures’ captions were updated to include the meaning of error bars and symbols. An Excel file with the data used to create the figures will also be included as supplementary material upon request.
With which statistical tests were the pairwise comparisons performed (Lin. 22) to state: "Pairwise comparisons showed 2 that men were faster in all decades"?
Answer: We used the least significant difference (LSD) post-hoc technique for the pairwise comparisons. This information was added to the Statistical Analysis section, as suggested in the next comment.
In the statistical section, they should describe in detail all the statistical tests used, including post hoc tests.
Answer: We agree with the expert reviewer. Thus, we added the tests used for normality, homogeneity, and post-hoc. Please see all changes marked as red.
Conclusions should be drawn on the basis of the data shown:
With the data shown you cannot state in the conclusion: "The greater improvement of women in cycling speed during the decades supports the hypothesis that women could outperform men in time-limited ultra-cycling races", perhaps you could state that the performances between men and women could be equalized.
Answer: We agree with the expert reviewer and changed to ‘The existing sex differences between men and women in cycling speed decreased with the duration of ultra-cycling races and with increasing age. Women cycled nearly as fast as men in the 24-hour races. The performance between men and women could be equalized because of the greater improvement of women in cycling speed during the decades time-limited ultra-cycling races. The improvement is shown in all age groups. The average speed of men and women started to converge in the 24-hour races. With the results of this study cyclists, sports medicine specialists and coaches could set long-term training goals. It would be of interest to examine the reasons for the greater performance improvement in women and the reasons for the better performances of women in longer lasting ultra-cycling races.
Round 2
Reviewer 1 Report
Rev-2
- According to the authors, the increase of athletes' efficiency is mainly influenced by the development of technology. Please mention the importance of exercise testing in the assessment of performance and predicting the training load. Development of sports biochemistry, sports dietetics or pharmacological support and development of wellness systems.--- The author mentioned only exercise tests
- The journal to which the paper is submitted includes publications on medical topics or the explanation of occurring phenomena is based on physiological-biochemical analysis. The work presented for review is rather of reporting nature and fits into the typical sports journals.----- not included in the manuscript information about the biological mechanisms of the described differences.
- Zapis literatury jest niezgodny z wymogami czasopisma; do poprawienia ---- nadal zawiera błędy
Author Response
Dear editor and referees,
thanks for your review.
Considering your comments, we made changes in the manuscript:
We added the biological reasons for the sex differences in the introduction and in the discussion.
Furthermore, we added improvements in performance, such as nutrition, recovery modalities and LLLT. And the technological improvements gain less space in the performance improvement.
We discarded the part with the sarcopenia and added muscle fiber and skeletal muscle mass differences between the sexes.
The topic does not contain the time period anymore.
And we have adapted the citation according to the specification.
Kind regards
The authors
Reviewer 3 Report
The document has been improved. But I still do not understand why the time periods that have been collected in each of the types of sports test (6, 12 and 24 hours) are different, when the aim is to compare the evolution over time of the differences in performance according to sex and age. At least they should remove "from 1983-2019" from the title, as it can be misleading. They should confine themselves to the data presented in Table 1. And likewise, Table 1 and the figures should represent the results from the same time periods, it should be from 2005-2021 (2005 is the lowest common denominator of the three types of sports competitions).
Sarcopenia is a pathological entity with clear diagnostic criteria (pathological loss of muscle mass accompanied by pathological loss of strength or performance). Sex differences in performance in endurance athletes can be argued with the difference in muscle mass between men and women or across the life span, but not with sarcopenia. People with sarcopenia cannot complete ultra-endurance tests. Lines 303-309 should be deleted.
Author Response

(The authors gave the same response as above.)
